# Evolving Artificial Intelligence (AI) at the Crossroads: Potentiating Productive vs. Declining Disruptive Cancer Research

**DOI:** 10.3390/cancers16213646

**Published:** 2024-10-29

**Authors:** Nilesh Kumar Sharma, Sachin C. Sarode

**Affiliations:** 1Cancer and Translational Research Lab, Dr. D.Y. Patil Biotechnology & Bioinformatics Institute, Dr. D.Y. Patil Vidyapeeth, Pune 411033, Maharashtra, India; 2Department of Oral Pathology and Microbiology, Dr. D.Y. Patil Dental College and Hospital, Dr. D.Y. Patil Vidyapeeth, Pune 411018, Maharashtra, India; drsachinsarode@gmail.com

**Keywords:** cancer cell, artificial intelligence, generative artificial intelligence, disruptive science, biological intelligence, health and diseases

## Abstract

In managing cancer diseases, emerging tools and technologies are pivotal for stakeholders including basic scientists, preclinicians, clinicians, and health data managers. In recent years, the emergence of artificial intelligence (AI)-based tools and programs has helped in significant ways to achieve holistic approaches including the diagnosis, therapy, and prognosis of cancer patients. It is true that in achieving enhanced management of cancer patients, productivity in terms of various forms of data, research publications, and patents centered on oncology has increased many-fold. Besides the positive impact of AI on cancer patients and involved stakeholders, reliable balanced approaches are also needed at the moment so that issues of ethics such as patient privacy and data bias can be resolved. Also, the future influence of AI on the creative and disruptive abilities of involved researchers is safeguarded by policies and guidelines on the extent of its uses in oncology.

## 1. Introduction

Artificial intelligence (AI), with domains such as artificial “general” intelligence (AGI) generative artificial intelligence (GenAI), and other forms of AI, comprises platforms considered crucial for understanding biological systems, the molecular basis of diseases, human behavior, and health data management, especially in the field of oncology [1,2,3,4,5]. AI is known for various attributes such as learning, reasoning, and problem-solving, which are also attributed to humans, enabling it to perform various tasks, including disruptive research in several facets of academia such as cancer research.

In recent times, diagnosis, therapeutic, and prognostic avenues in cancer research have seen the integration of AI platforms such as AGI and GenAI, which have made significant contributions to the management of cancer patients [6,7,8,9,10,11]. Such integration of AI platforms such as AGI and GenAI has led to productive cancer research in terms of output parameters such as number of publications and citations over time [12,13,14,15].

GenAI-based chatbots tools like ChatGPT have assisted preclinical and clinical scientists in extracting meaningful molecular and cellular information, contributing to productive cancer research, and have in turn been beneficial to cancer patients [16,17,18,19,20,21]. Furthermore, drug repurposing, precision oncology, and management of cancer research data at a large scale are driven by various AI platforms such as AGI and GenAI.

Besides the necessary implications of AGI and GenAI in productive cancer research, concerns are pertinent about their potential influence on disruptive sciences and innovations that have happened in the last several decades, such as the discovery of various hallmarks such as the molecular basis of cancer, immune regulation in cancer, and CAR T cell anticancer therapies [17,18,19,20,21,22,23,24,25]. Therefore, there is a need for a cohesive and consensus-based approach to create a balance between AI-driven research productivity and the emergence of human intellectual capabilities that drive disruptive cancer research. This perspective review highlights both sides of AI in the context of cancer research: that which can accelerate productive research and that which might potentially hinder disruptive cancer research.

## 2. Artificial Intelligence

Artificial intelligence (AI) refers to computer programs that can perform learning, reasoning, interventional tasks to solve problems, thinking, making predictions, and abstracting meaningful information from large data sets and information, similar to human intelligence [1,2]. 

AI tools in the context of biological systems, human behaviors, and handling of health data on diseases including cancer are known as GenAI [3,4,5]. On the other hand, AGI is attributed to AI tools acting as theoretical representations of generalized human cognitive, abstraction, creative, reasoning, and comprehension abilities to perform problem-solving tasks. AGI is considered a subset of AI as a theoretical field that is intended for the development of machines that are attributed to human-like general intelligence. Some notable examples of AGI are virtual assistants (e.g., Alexa, Santa Barbara, CA, USA, Cortana), deep learning, and large multimodal models (LMMs). AGI is predicted to possess human-like intellect in its ability to comprehend, learn, and mimic human cognitive skills [3,4,5].

Another approach to AI, known as GenAI, has evolved to create new content within specific domains with prompt intervention by human input. In other words, the abilities of GenAI are dependent on the learning and cognitive abilities of humans and human-matched AI tools such as AGI [5,6,7]. GenAI is commonly used to comprehend large sets of content such as texts and data to create new content, similar to human capabilities [5,6,7]. ChatGPT and other LLMs are represented as tools that use existing data sets to create new forms of texts, images, and data sets. GenAI, a type of machine learning, is usually known to exist in the form of chatbots such as ChatGPT and LLMs that can generate new human-comprehended text, images, audio, video, and code from existing texts, data, and images from various knowledge domains including oncology [7,8,9]. 

ChatGPT is known as an AI chatbot that can generate human-like text by using forms of LLMs. Deep learning is known to encompass neural networks that can guide us to a promising path of AGI [7,8,9]. At the same time, LLMs are not considered a form of AGI, but rather can be the initial step towards AGI. LLMs are a type of AI that focuses on understanding and generating human language. LLMs are capable of processing and creating text [5,6,7,8,9]. In contrast, AGI is aspiring and evolving to match human-like intelligence. 

Both GenAI and AGI will continue to evolve with the help of LLMs, machine learning (ML), and deep learning (DL) algorithms. Here, ML is defined as a subfield of AI that can learn from data and perform tasks using algorithms, while Dl is a subset of ML that learns from a large amount of data using neural networks [7,8,9]. In the approach by AI to mimic human intelligence, fuzzy logic and neural networks are an integral part of its methodologies. Also, AGI involves various approaches including the data-driven AI (DAI) paradigm and the symbolic AI (SAI) paradigm, which are guided by human-based cognitive and comprehension abilities [1,2,3,4,5,6,7,8,9].

AI tools in the form of large language models provide timely and accurate extraction of knowledge from a vast pool of research publications amounting to more than five million with cancer as a keyword according to search engines such as Pubmed, Scopus, and Web of Sciences. Processing of images from biopsies by AI tools could minimize the burdens on clinical doctors so that they can better focus on the important diagnostic and therapeutic modalities for cancer patients. 

## 3. Potentiating Productive Cancer Science

The evolving abilities of AI applications in the forms of ChatGPT and MetaAI that use LLMs are helping preclinical and clinical scientists extract meaningful formation at the molecular, intracellular and intercellular levels from tissue attributes in all dimensions including geometrical, physical, and mechanical landscapes. It is widely accepted that AI tools, in the hands of preclinical and clinical scientists, sometimes uncover unnoticeable and masked molecular and cellular information from tumor samples, contributing to the diagnostic, therapeutic, and prognostic management of cancer patients [10,11,12,13,14,15]. 

A subset of AI, predictive AI models, is being developed to assist preclinical and clinical doctors in predicting the success, failure, and side effects of various cancer therapies, including conventional chemotherapy, radiation, surgery, and precision-guided targeted therapies [12,13]. In the future, a project on AI with the capability to predict the timing of such cancer therapies for cancer patients, for example to predict whether surgery can be performed on a cancer patient in the morning or evening, should be developed. Predictive AI could be trained from a large preclinical and clinical data set so that it can suggest a combination of dietary components for cancer patients. Even AGI-driven tools are driving the success of drug repurposing for cancer patients and could contribute to productive cancer research [15,16,17].

It is encouraging to witness the uses of AI tools in helping to identify tumor heterogeneity at different levels, including body systems, organ systems, tissue systems, cells, molecules, and even electron heterogeneity. In the future, the data collected from all such heterogeneity and their analysis by AI tools could pave the way for a better understanding of predictive therapies for cancer patients [12,13,14,15,16,17]. 

The scope of AGI lies in strengthening the core part of research as well as medical devices and instruments that can help preclinicians and clinicians comprehend and manage large research and clinical data sets. This impetus, driven by AGI and other forms of intelligent systems, is facilitating the productive sciences. 

AI-supported data repositories such as data on clinical trials, international agencies’ data sources, and hospital-generated data sources will converge so that preclinicians and clinicians can access the data promptly and with greater accuracy. Even genomic and molecular databases such as The Cancer Genome Atlas (TCGA) are well supported by AGI and other intelligent support approaches, helping make cancer research more productive in terms of both the number of publications and the creation of large datasets [13,14,15,16,17]. AGI tools, in the forms of various facilitative databases, repositories, clinical image processing, molecular data handling, and content generation for research publications, are well supported by the current data collected from NCBI (PUBMED) on keywords related to “Cancer” [17,18,19,20,21,22,23,24]. A summary of AGI, GenAI, and other AI-assisted platforms that contribute to productive cancer research is presented (Figure 1).

The total number of publications in the last 100 years as of the date of writing (30 September 2024) is 5 million. Almost half of them, more than 2 million research publications, were produced in the last ten years and more than 2 million were produced in the last five years [20,21,22,23,24,25]. This volume of productive research publications on cancer research was undoubtedly fueled by various forms of GenAI-based tools such as ChatGPT and LLMs. 

In recent developments, the emergence of AI tools with capabilities to mine data even at the individual cancer cell level, among various cellular populations in the tumor niche and other cellular hallmarks, along with prediction intelligence on the success and failure of specific drugs, is paving the way for intelligent precision oncology for existing and future cancer patients [22,23,24,25]. Recently, the emergence of cloud AI platforms driven by ML and DL approaches has helped researchers and clinicians achieve an astounding level of collaboration, enabling access to large data sets related to molecular, cellular, and clinical findings, which may have contributed to the large increase in research publications. Besides the additive role of AGIs and other intelligent tools and platforms, several research publications’ websites, databases, content creation systems, and authentications of data, tests, and images are driven by GenAI-based tools such as ChatGPT and LLMs, which as, in turn, increased the number of productive cancer research publications. 

The uses of AI in various domains of oncology including the diagnosis, precision therapy, and prognosis of cancer patients may face concerns of various forms of ethics such as patient privacy, autonomy and potential bias, as well as limitations to the accuracy of data interpretations. Therefore, in an era of productive cancer science with enormous research publications, research data and various databases on oncology, guidelines and policies should be in place to safeguard the ethics issues centered on the privacy and safety of cancer patients.

## 4. Declining Trends of Disruptive Sciences

In the last century, the milestone discoveries in disruptive sciences in cancer research have made significant contributions for preclinical scientists, clinicians, cancer patients, and society at large by revealing various tumor hallmarks [17,18,19]. A better understanding of notable tumor hallmarks, such as sustained proliferative signaling, evading growth suppressors, resisting cell death, enabling replicative immortality, inducing angiogenesis, activating invasion and metastasis, reprogramming energy metabolism, evading immune destruction, tumor-promoting inflammation, gain of functions of proto-oncogenes, and loss of functions of tumor suppressor genes, has been possible due to such outstanding disruptive sciences in cancer research [17,18,19,20,21,22,23,24,25]. In this remarkable journey of cancer research by outstanding scientists, we have achieved appreciable progress in the diagnostic, therapeutic, and prognostic management of cancer patients. 

Some notable disruptive discoveries that have shaped the key understanding of the molecular basis of cancer include the discovery of the first proto-oncogene tyrosine-protein kinase Src and the first human rat sarcoma oncogene by Dr. Robert Weinberg [17,18,19,20,21,22]. Another disruptive discovery proposed the relevance of checkpoint inhibitors harnessing the immune system to combat cancer [19]. In the past decade, the discovery of PD-1 has formed the basis of the development of immunotherapy drugs such as pembrolizumab [20,21]. To understand the origin of cancer, the identification of cancer stem cells as the seeds of tumor imitation has emerged as one of the compelling bases of cancer [22]. 

The invention of next-generation sequencing (NGS) and the Human Genome Project significantly contributed to achieving disruptive research in cancer science [23]. Another milestone in disruptive science includes the discovery of epigenetic regulation in cancer cells, which contributes to various hallmarks such as drug resistance, through notable enzymes such as histone methyltransferases [24]. Recently, the discoveries of CAR-T cell and adoptive T cell therapies have been milestones in disruptive cancer research [25,26]. Emerging tumor hallmarks such as metabolic reprogramming and the tumor microbiome are playing a significant role in improving the therapeutic and prognostic management of cancer [17,18,19,20,21,22,23,24,25]. In summary, the above-highlighted examples of disruptive sciences provide appreciable pieces of evidence on the timelines before the advent of AI and ML-assisted tools and technologies. There is no ambiguity that AI- and ML-assisted tools and technologies are making substantial transformations in the lives of all stakeholders of cancer research including preclinicians, clinicians, and cancer patients to achieve holistic management of cancer in the future.

As discussed above, despite the impactful contribution of AGI tools in the potentiation of productive cancer research, a critical discussion on the influence of AGI on disruptive cancer science is reasonable. Disruptive sciences including those in the cancer field have considered landmark discoveries and innovations that have improved the health and life of cancer patients [17,24,25,26,27]. The decline in disruptive sciences that spur novel discoveries is debated, and various reasons are proposed, including the working environment, infrastructure, human resources, and policies by concerned stakeholders, including the government [20,21,22,23,24,25,26,27,28].

At the same time, the discovery of artificial intelligence (AI) decades ago coincided with the declining trends of disruptive sciences [18,19,20,21,22,23,24,25,26,27,28,29]. Hence, it would be logical to debate and critically propose viewpoints on whether or not AI could be one of the potential factors that may have influenced the disruption in the research on disruptive sciences. For example, milestone discoveries in cancer research, including discoveries of oncogenes, immune hallmarks, and anticancer drugs such as cisplatin and doxorubicin, happened almost a decade ago. At the same time, research publications in the last 10 years account for almost half of the total publications in cancer sciences in the last 100 years. A flow model on the possible influences of AGI, GenAI, and other AI platforms on the decline in disruptive cancer research is presented in Figure 2.

In health science and technology, including preclinical and clinical oncology, the biological intelligence of scientists in terms of abstraction, creativity, critical thinking, writing, and many more may drive novel discoveries, starting from oncogenes to gene editing systems. However, it is highly pertinent to ask whether or not interferences by AI tools such as ChatGPT could influence the abstraction and creative thinking processes of scientists. 

In turn, AI tools such as ChatGPT could lead to a redundancy of the biological intelligence of scientists, and hence cause a disruption in abstraction, creativity, and critical thinking, which could contribute to a decline in the disruptive sciences. 

Besides comprehension, abstraction, creativity, and critical thinking, the use of ChatGPT, and other AI tools reduces the use of biological intelligence in terms of interpretation, inferences, and analytical thinking when dealing with various images and data. Thus, the AI-driven ChatGPT could compromise all these attributes related to the biological intelligence of scientists, and such evolutionary adaptations and learning by AI will further hinder research in the disruptive sciences.

In the current scenario of productive cancer research publications, the translational impact of such publications in the form of new drug development and clinical outcomes is also debatable [30,31,32,33,34,35,36]. Preclinical and clinical scientists use AGI tools such as ChatGPT as assistive and productive tools. However, prolonged reliance on ChatGPT and other OpenAI tools over a generation could also lead to genetic and epigenetic modulations in human biological cognitive intelligence that could be inheritable by the next generations. 

## 5. Futuristic Implications

The integration of AI in oncology has touched upon various facets of cancer research, including the diagnosis, treatment, prognosis, and therapy management of cancer patients. GenAI-based chatbots such as ChatGPT and LLMs have contributed to productive cancer research in terms of the millions of publications, the creation of databases, websites, cloud platforms, and the establishment of many more additive systems for preclinicians, clinicians, professionals in the management of cancer patients, as well as for organizations including governments and NGOs. However, the positives and limitations are important to discuss regarding their impact on disruptive sciences in cancer research.

Several positive impacts of AI tools are notable, such as AI-driven personalized treatment strategies, the facilitation and acceleration of rapid data analysis by AGI, and the handling of redundant and repetitive assignments by various AGI programs. Other positives include AI-assisted diagnosis and treatment for the better survival of cancer patients and global research collaboration facilitated by cloud-based AI platforms. It is widely accepted that existing GenAI and future AI programs will allow for breakthroughs in the lives of cancer patients through futuristic humanoid robots with capabilities equivalent to those provided human intelligence that can perform diagnoses, make therapeutic decisions, and deliver other forms of cancer management services for patients. 

Besides the abovementioned positive and promising implications of AGI, GenAI, and other AI platforms, emphasis should be placed on understanding how the decline in disruptive sciences in cancer research can be mitigated. Such issues can be improved by establishing certain guidelines for preclinicians and clinicians regarding the modalities in using GenAI, and other AI platforms. This ensures that human intelligence can contribute to future disruptive discoveries through the direct interventions of human cognitive abilities.

In other words, preclinicians and clinicians may be encouraged to manage over-reliance on AI platforms, which may otherwise significantly impact human creativity and innovations in the long term in the journey of disruptive research. In this context, attention is needed to understand how the prolonged use of AI tools could influence long-term genetic and epigenetic modulations in human cognitive intelligence. Biological intelligence is associated with analytical thinking, and hence AI-driven analysis of cancer research data by preclinicians and clinicians could negatively impact human intelligence.

Thorough development of AI programs to handle large amounts of data in the forms of texts, images, and the medical records of cancer patients would be crucial too for handling data privacy and security, which should be addressed in future research.

Positive discussion on all the possible impacts of AI on scientific trends will be appreciated. Regarding the experimental pieces of evidence, it would be fascinating and empirical to investigate the impact of AI at molecular levels that could be directly or indirectly associated with a decline in the disruptive sciences through prospective studies in the coming years.

In the future, holistic approaches are needed to achieve a balanced integration of GenAI and other AI platforms with human intelligence so that productive as well as disruptive science research will be fostered for cancer patients and society at large. AGI, GenAI, and other AI platforms could be trained, taught to learn, and developed in such a way that could lead to the enhancement of human cognitive abilities and at the same time save time via the use of AI-assisted inputs for researchers’ redundant and repetitive tasks. Hence, a future of human and AI symbiotic platforms as well as any biological discoveries may lead to productive and disruptive cancer research.

## 6. Conclusions

In summary, the various forms of AI including ChatGPT, ML, and deep learning influence productive and meaningful research publications. On preclinical and clinical platforms, AI tools are modulating cancer patients’ diagnostic, therapeutic, and prognostic outcomes in various landscapes including databases, image processing, and the comprehension of large tests and data generated in preclinical and clinical settings related to cancer. At the same time, the emergence and excessive intervention of AI tools such as ChatGPT could also influence human cognitive abilities, contributing to disruptive and discovery-driven research in cancer science. Therefore, a balance between AGI-based and human intelligence needs to be emphasized, so that productive science and disruptive science can work in parallel for the benefit of cancer patients and society.

## Figures and Tables

**Figure 1 cancers-16-03646-f001:**
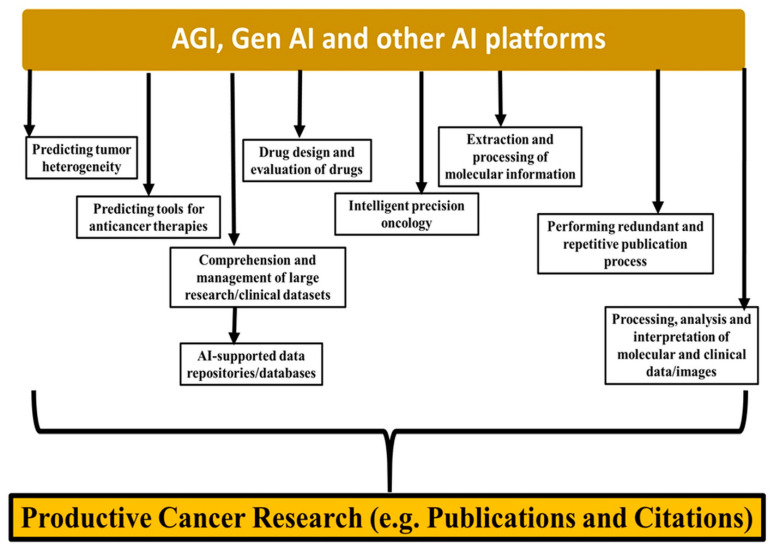
AGI, GenAI, and other AI platforms that contribute to enhancing productive cancer research in terms of the magnitude and number of publications and citations.

**Figure 2 cancers-16-03646-f002:**
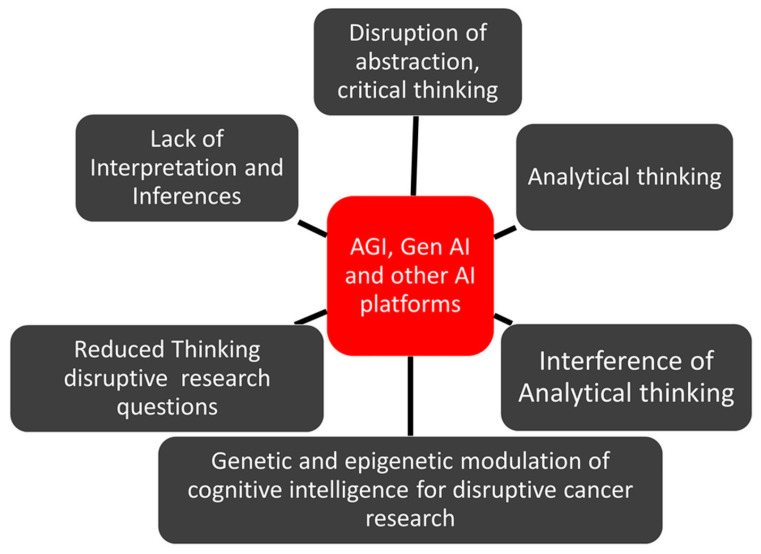
A flow model on the influence of AGI, GenAI, and other AI platforms on various attributes of human intelligence contributing to disruptive cancer research.

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
