# Peer review of "Evolving Artificial Intelligence (AI) at the Crossroads: Potentiating Productive vs. Declining Disruptive Cancer Research"

_cancers, 2024, doi:10.3390/cancers16213646_

Round 1

Reviewer 1 Report

Comments and Suggestions for Authors

This study presents a perspective on the dual role of AI in cancer research. While it highlights the benefits of AI in enhancing research productivity, it raises concerns about its potential to disrupt traditional scientific methods and creativity. 

The authors provide a thorough examination of both artificial general intelligence (AGI) and generative AI (GenAI), detailing their applications in cancer research. This dual focus is valuable for understanding the AI technologies in oncology. AI tools like AGI and GenAI streamline data analysis, allowing researchers to focus on more complex problems. This can lead to increased research output and efficiency.

Here are several comments regarding the paper.

Page 2, line 66-93. The paper does not provide details on how AI handles data privacy and security in cancer research. What are the ethical considerations surrounding AI-driven cancer research?

Page 2, line 77. What is LLM? Please write in full before using the abbreviation.

Page 2, line 84. Please write ML in full before using the abbreviation.

Page 3, line 91. Why only Pubmed and Scopus? How about Web of Science?

Page 5, line 195. References 28-20?

Page 5, line 167&168. Please write SRC and HRAS in full before using the abbreviation.

Page 7, line 264. large. In the future, 264 AGI, gen AI, and other AI platforms could be trained,… the words “In the future” has been used in page 7, line 262. Redundant.

Certain phrases and ideas are repeated throughout the text without adding new insights, which can lead to redundancy and dilute the overall message. This repetition could confuse readers and detract from key points.

Overall, the arguments regarding the decline of disruptive sciences due to AI reliance are largely speculative. The paper makes broad claims about AI's negative impact on disruptive sciences without providing empirical data or case studies to support these statements. This undermines the credibility of the arguments presented. 

The authors had also overgeneralized that AI tools could lead to a decline in biological intelligence. The paper should consider counterarguments or examples where AI has enhanced rather than hindered creativity. 

Comments on the Quality of English Language

Moderate editing is needed

Author Response

Reviewer 1.

General Comment: This study presents a perspective on the dual role of AI in cancer research. While it highlights the benefits of AI in enhancing research productivity, it raises concerns about its potential to disrupt traditional scientific methods and creativity. The authors provide a thorough examination of both artificial general intelligence (AGI) and generative AI (GenAI), detailing their applications in cancer research. This dual focus is valuable for understanding the AI technologies in oncology. AI tools like AGI and GenAI streamline data analysis, allowing researchers to focus on more complex problems. This can lead to increased research output and efficiency.

Response: The authors appreciate constructive viewpoints on intersections between AI tools and research productivity and at the same time potential implications in declining disruptive sciences.

Here are several comments regarding the paper.

Comment 1: Page 2, line 66-93. The paper does not provide details on how AI handles data privacy and security in cancer research. What are the ethical considerations surrounding AI-driven cancer research?

Response 1: A paragraph on ethical considerations is provided. The uses of AI in various domains of oncology including diagnostic, precision therapeutic, and prognosis of cancer patients may face concerns of various forms of ethics such as patient privacy, autonomy, and potential bias and limitation at the accuracy of the data interpretation. Therefore, in a time of productive cancer sciences with enormous research publications, research data and various databases on oncology, guidelines, and policies should be in place to safeguard the ethics issues centered on the privacy and safety of cancer patients. "A robust development of AI programs to handle large amounts of data in the forms of texts, images, and medical records of cancer patients would be crucial too, for handling data privacy and security" is added in the futuristic sections.

Comment 2: Page 2, line 77. What is LLM? Please write in full before using the abbreviation.

Response 2: The authors have included a full form of large language models (LLMs)  in the texts

Comment 3: Page 2, line 84. Please write ML in full before using the abbreviation.

Response 3: The authors have included a full form of machine learning (ML) in the texts

Comment 4: Page 3, line 91. Why only Pubmed and Scopus? How about Web of Science?

Response 4. The authors have included Web of Science.

Comment 5.  Page 5, line 195. References 28-20?

Response 5: Corrected as (20-28)

Comment 6: Page 5, line 167&168. Please write SRC and HRAS in full before using the abbreviation.

Response 6: The authors have included expanded form. “discovery of the first proto-oncogene tyrosine-protein kinase Src and the first human oncogene, human rat sarcoma oncogene)

Comment 7: Page 7, line 264. large. In the future, 264 AGI, gen AI, and other AI platforms could be trained,… the words “In the future” has been used in page 7, line 262. Redundant.

Response 7: The authors appreciate the suggestion and modified the texts.

General comment 1: Certain phrases and ideas are repeated throughout the text without adding new insights, which can lead to redundancy and dilute the overall message. This repetition could confuse readers and detract from key points.

Response: The authors appreciate constructive suggestions. The authors have added more clarity and uniformity to AGI and GenAI so that readers will have better comprehension.

General comment 2: Overall, the arguments regarding the decline of disruptive sciences due to AI reliance are largely speculative. The paper makes broad claims about AI's negative impact on disruptive sciences without providing empirical data or case studies to support these statements. This undermines the credibility of the arguments presented.

Response 2: The authors consider comments as a positive discussion on the how and what extent of impact various forms of AI may have in the decline of disruptive sciences. The authors have extended their views based on the trends of disruptive discoveries in the last decade and speculated how they may be extrapolated in the future. Viewpoints are emerging that could link the uses of AI tools and trends of disruptive sciences. (Prillaman M. Is ChatGPT making scientists hyper-productive? The highs and lows of using AI. Nature. 2024. 627(8002):16-17.

Conroy G. How ChatGPT and other AI tools could disrupt scientific publishing. Nature. 2023. 622(7982):234-236.

Park M, Leahey E, Funk RJ. Papers and patents are becoming less disruptive over time. Nature. 2023. 613(7942):138-144. Castelvecchi D, Callaway E, Kwon D. AI comes to the Nobels: double win sparks debate about scientific fields. Nature. 2024 Oct 10. doi: 10.1038/d41586-024-03310-8.

At the same time, positive discussion in all possibilities on the impact of AI on scientific trends should be welcome. Regarding the experimental pieces of evidence, the authors think that it would be too early to see the right experimental design at molecular levels about how AI may modulate these including genetic and epigenetic signatures in human cells. At the same time, future studies would be fascinating and empirical to investigate the impact of AI at molecular levels that could be directly or indirectly associated with decline in the disruptive sciences.

General comment 3: The authors had also overgeneralized that AI tools could lead to a decline in biological intelligence. The paper should consider counterarguments or examples where AI has enhanced rather than hindered creativity.

Response 3: The authors truly agree with such viewpoints that besides the impact of AI tools and programs in the downward trend of disruptive sciences, an upward influence on

Creativity and disruptive intelligence are a possibility. In such a scenario, it would be too early to collect experimental data and we should encourage to design of such promising experiments that can provide data that can be helpful in the better understating of the impact of AI in disruptive sciences in both perspectives.

Overall, the authors truly appreciate impactful suggestions by learned experts. The authors also acknowledge that the perspectives on the positive and negative influences of AI on productive sciences and disruptive sciences are emerging and presented viewpoints may invite further discussion for a better future for all stakeholders including scientists and cancer patients.

Reviewer 2 Report

Comments and Suggestions for Authors

This article is presenting the divide between the increased number of scientific publications in the cancer area and the insignificant discoveries leading to impactful outcomes. The argument is a hard one to prove given the wide range of cancer and deep studies in each of its domains. So stabilizing the thesis demands a heavy study and rigorous stats overcasting these domains which before that is rendering this claim rather an arbitrary one albeit seemingly true. Apart from this, the presentation style is having a great leeway to improve. The terms are not perfectly crafted in their definition and used pretty loosely. The narration is also suffering from multiple repetitions which is making understanding the main thesis of the paper more difficult. I think also the scope of the thesis could be much more expanded with providing more biological examples to support the main finding of this article as summarized in the figures. 

More specifically here are some points to consider:

L17: What is AGI?

L40: Is it GAI or AGI?

L52: I was under impression the ChatGPT is a GenAI?

L56-57? Space between Gen” “AI.

L71: general artificial intelligence -> GAI?

L82: Achieve is a transitive verb needing an object!

L89: In L77 LLM is used? Why not here?

L95: is it finally General or Generalized or it does matter?

L147: clinicians and clinicians?

Author Response

Reviewer 2:

General comment: This article is presenting the divide between the increased number of scientific publications in the cancer area and the insignificant discoveries leading to impactful outcomes. The argument is a hard one to prove given the wide range of cancer and deep studies in each of its domains. So stabilizing the thesis demands a heavy study and rigorous stats overcasting these domains which before that is rendering this claim rather an arbitrary one albeit seemingly true. Apart from this, the presentation style is having a great leeway to improve. The terms are not perfectly crafted in their definition and used pretty loosely. The narration is also suffering from multiple repetitions which is making understanding the main thesis of the paper more difficult. I think also the scope of the thesis could be much more expanded with providing more biological examples to support the main finding of this article as summarized in the figures. 

Response: The authors appreciate constructive viewpoints. There is no doubt that AI tools and technologies have contributed to productivity in terms of publications and AI-assisted discoveries. In recent, milestone recognitions are visible with Nobel prizes in Physics and Chemistry that see the interface between AI and Biological sciences. It is also true that it would be hard to provide direct evidence or data rather than indirect evidence in terms of an increase in the number of publications on cancer and potential observations of disruptive discoveries. The intent behind this perspective is to invite constructive deliberations and insights so that AI-assisted cancer sciences may retain balances between productive and disruptive sciences.

More specifically here are some points to consider:

Comment 1: L17: What is AGI?

Response 1: The authors appreciate remark. Here general artificial intelligence (AGI) is intended word. A correction is inserted in the text.

Comment 2: L40: Is it GAI or AGI?

Response 2: Generative artificial intelligence abbreviated as (GenAI) in this paper and, otherwise also called GAI. On the other hand, General Artificial Intelligence is abbreviated as AGI in this paper.

Comment L52: I was under impression the ChatGPT is a GenAI?

Response: The authors have corrected the nomenclature with better Uniformity on AGI and GenAI. The learned reviewer is right that ChatGPT is a form of application of GenAI that uses LLMs. Sometimes, LLMs are suggested as an initial step towards developing AGI that potentially mimic human intelligence abilities.

Comment L56-57? Space between Gen” “AI.

Response: The authors appreciate observation. We have made uniformity for GenAI and made changes.

Comment L71: general artificial intelligence -> GAI?

Response: The authors have revisited the nomenclature and corrected it with Uniformity as Artificial General Intelligence (AGI) and Generative artificial intelligence (GenAI).

Comment L82: Achieve is a transitive verb needing an object!

Response: The authors have revised the sentence.””In contrast, AGI is aspiring and evolving to match with human-like intelligence”.

Comment L89: In L77 LLM is used? Why not here?

Response. The authors appreciate valuable suggestions and points are included in the texts.

Comment L95: is it finally General or Generalized or it does matter?

Response: The authors express appreciation for meticulous observations. We have made a better uniformity across the manuscript for AGI and GenAI.

Comment: L147: clinicians and clinicians?

Response: The texts are changed with preclinicians and clinicians

Overall, the authors truly acknowledge impactful suggestions by learned experts. The authors also acknowledge that the perspectives on the positive and negative influences of AI on productive sciences and disruptive sciences are emerging and presented viewpoints may invite further discussion for a better future for all stakeholders including scientists and cancer patients.

Reviewer 3 Report

Comments and Suggestions for Authors

The main theme of the present perspective seems interesting, is a good read and much needed in the present times of AI era. Following are some minor comments that need to be addressed:

1.      Typographical and grammatical corrections are must for the whole content; instance in the abstract line 17, correct AGI as artificial ‘general’ intelligence.

2.      Write generative AI (GenAI) in a uniform way with or without space in between. Please check the whole manuscript.

3.      Briefly explain the difference between Artificial intelligence, Machine Learning and Deep Learning.

4.      Explain clearly with reference citations the genetic and epigenetic modulations in human biological cognitive intelligence and their potential disruption by AI.

5.      Mention the date when the literature databases PubMed and Scopus were accessed.

6.      Page 2, Line 77, expand LLMs.

Author Response

Reviewer 3:

General Comment: The main theme of the present perspective seems interesting, is a good read and much needed in the present times of AI era. Following are some minor comments that need to be addressed:

Response: The authors appreciate time and viewpoints on the perspective.

Specific comments

    Comment 1.      Typographical and grammatical corrections are must for the whole content; instance in the abstract line 17, correct AGI as artificial ‘general’ intelligence.

      Response 1: The authors are thankful for important suggestions. The texts are corrected and made uniform as artificial ‘general’ intelligence (AGI).

Comment 2.      Write generative AI (GenAI) in a uniform way with or without space in between. Please check the whole manuscript

Response 2: The authors appreciate suggestions and appropriate corrections included generative artificial intelligence (GenAI)

Comment 3.      Briefly explain the difference between Artificial intelligence, Machine Learning and Deep Learning.

Response 3: Explanations about Machine Learning and Deep Learning are provided.

Comment 4.      Explain clearly with reference citations the genetic and epigenetic modulations in human biological cognitive intelligence and their potential disruption by AI.

Response 4: The authors consider comments as a positive discussion on the how and what extent of impact various forms of AI may have in the decline of disruptive sciences. The authors have extended their views based on the trends of disruptive discoveries in the last decade and speculated how they may be extrapolated in the future. Viewpoints are emerging that could link the uses of AI tools and trends of disruptive sciences. (Prillaman M. Is ChatGPT making scientists hyper-productive? The highs and lows of using AI. Nature. 2024. 627(8002):16-17.

Conroy G. How ChatGPT and other AI tools could disrupt scientific publishing. Nature. 2023. 622(7982):234-236.

Park M, Leahey E, Funk RJ. Papers and patents are becoming less disruptive over time. Nature. 2023. 613(7942):138-144. Castelvecchi D, Callaway E, Kwon D. AI comes to the Nobels: double win sparks debate about scientific fields. Nature. 2024 Oct 10. doi: 10.1038/d41586-024-03310-8.

At the same time, positive discussion in all possibilities on the impact of AI on scientific trends should be welcome. Regarding the experimental evidence, the authors think that it would be too early to see the right experimental design at molecular levels with regard to how AI may modulate these including genetic and epigenetic signatures in human cells. At the same time, future studies would be fascinating and empirical to investigate the impact of AI at molecular levels that could be directly or indirectly associated with decline in the disruptive sciences.

Comment 5.      Mention the date when the literature databases PubMed and Scopus were accessed.

Response 5: The date is included.

Comment 6.      Page 2, Line 77, expand LLMs.

Response 6: Expanded forms of large language models (LLMs) are included in the texts.

Overall, the authors truly acknowledge impactful suggestions by learned experts. The authors also acknowledge that the perspectives on the positive and negative influences of AI on productive sciences and disruptive sciences are emerging and presented viewpoints may invite further discussion for a better future for all stakeholders including scientists and cancer patients.

Round 2

Reviewer 2 Report

Comments and Suggestions for Authors

Thank you for your updates and responses to my comment. The paper reads much better now.